# Determinants for late presentation of glaucoma among adult glaucomatous patients in University of Gondar Comprehensive Specialized Hospital. *Case-control study*

**Biruktayit Kefyalew Belete**[1], **Natnael Lakachew Assefa**[2]*, **Abel Sinshaw Assem**[2], **Fisseha Admasu Ayele**[3]

**1** Department of Ophthalmology and Optometry, School of Medicine, College of Medicine and Health Sciences, Hawassa University, Hawassa, Ethiopia, **2** Department of of Optometry, School of Medicine, College of Medicine and Health Sciences and Comprehensive Specialized Hospital, University of Gondar, Gondar, Ethiopia, **3** Department of of Ophthalmology, School of Medicine, College of Medicine and Health Sciences and Comprehensive Specialized Hospital, University of Gondar, Gondar, Ethiopia

☯ These authors contributed equally to this work.

\* natiuog@gmail.com

**Data Availability Statement:** All relevant data are within the paper and its Supporting Information files.

## Abstract

### Introduction

Glaucoma is a disease which causes optic nerve damage and remains a major public health concern worldwide. Late presentation is a major risk factor for glaucoma induced blindness. The aim of this study was to assess determinants for late presentation of glaucoma among adult glaucomatous patients.

### Methods

A hospital-based case-control study was conducted among 452 adult glaucomatous patients. Late presenters were glaucoma patients diagnosed with cup to disc ratio (CDR) > 0.8 and mean deviation of greater than -14 decibel in either of the eyes at their first presentation. Study participants were selected among glaucomatous follow-up patients by using systematic random sampling. Data were entered into EPI Info version 7 and exported to SPSS version 22 for analysis. Bivariable and multivariable logistic regression analysis was done to identify determinants. Variables with P-value < 0.05 were considered as statistically significant.

### Result

The mean age of participants were 55.1 ± 13.2 years. Being > 60 years of age, 4.51 times (AOR: 4.51; 95% CI: 1.74, 11.67), resided > 53 km away from the hospital 6.02 times (AOR: 6.02; 2.76, 13.14), Presenting IOP > 30 mmHg, 4.49 times (AOR: 4.49, 95% CI: 2.10, 9.12), poor knowledge of glaucoma, 4.46 times (AOR: 4.46, CI: 2.62, 7.58) and absence of regular eye checkup, 2.35 times (AOR: 2.35, 95% CI: 1.09, 5.47) higher odds of being late presenter.

**Funding:** The authors received no specific funding for this work.

**Competing interests:** The authors have declared that no competing interests exist.

## Conclusion

Increasing age, high IOP, poor knowledge of glaucoma, residing far away from the hospital and absence of regular eye checkups were significantly associated with late presentation.

## Introduction

Globally, glaucoma causes irreversible blindness in 4.6–6.7 million people [1] with a prevalence of 3.54% for the population aged 40 to 80 years. In 2013, the number of people aged 40 to 80 years with glaucoma was 64.3 million which was predicted to increase to 76.0 million in 2020 and 111.8 million in 2040 [2]. Glaucoma mainly affects developing nations, and Africa accounts for 15% of the world's blindness due to glaucoma [2]. Population-based studies in Asian countries showed a higher prevalence of glaucoma [3] and Primary Open Angle Glaucoma (POAG) is the most commonly reported [4–6]. The prevalence of previously undiagnosed glaucoma in South Africa was 87.0% [7]. In Ethiopia, glaucoma results an irreversible vision loss in 62,000 individuals, becoming the fifth common cause of blindness in the country [8]. In a study done in 2002 in North Shoa of Ethiopia, glaucoma accounted 11.4% of blindness [9].

Several studies estimated that 10–33% of people with glaucoma had advanced disease and visually impaired at the first diagnosis due to their late presentation [10–12]. The reason for the late presentation was due to lack of early symptoms [13,14], slowly progressive and asymptomatic of nature of glaucoma [15]. Previous studies showed age, sex, educational level, occupational group, poor socioeconomic status, high intraocular pressure (IOP) at presentation, pseudo-exfoliation, awareness and knowledge about glaucoma, absence of a positive family history of glaucoma as determinants for late presentation of glaucoma [16–19]. Other studies also showed that late presentation is a major risk factor for blindness due to glaucoma [17,19,20–22]. It has been estimated that in Africa, half of patients with glaucoma are blind in at least one of their eyes at presentation [23]. Glaucoma was one of the leading cause of irreversible blindness in Ethiopia [8], thus this study has an immense importance to salvage the community from glaucoma induced blindness. Previous published evidences in Ethiopia didn't have enough information to explore the determinant factors for the late presentation of glaucoma. Most of these studies were done to assess and the prevalence of blindness due to glaucoma [8,9] and the associated factors of glaucoma [24]. Because of this, it needs explicit in situ study to identify the determinants for late presentation of glaucoma. In addition, the result of this study will provide base line information for the health care workers, researchers, health care planers, policy makers and other stakeholders accordingly. Therefore, this study was aimed to assess determinants for late presentation of glaucoma among adult glaucomatous patients in Ethiopia.

## Materials and methods

### Study design, setting and sampling

A hospital-based case-control study was conducted in the University of Gondar, Comprehensive Specialized Hospital, tertiary eye care and training center (UoG CSH TECTC) Northwest Ethiopia, 2020. Gondar city is located 738 kilometers away from Addis Ababa, the capital city of Ethiopia. UoG CSH TECTC is the only tertiary eye care center in the city providing comprehensive eye care for the Northwest Ethiopia which provides services for glaucoma patients from early diagnosis to frequent follow-up.

All adult glaucomatous patients aged $\geq$ 18 years, diagnosed within the last two years and on follow-up were included in the study. Whereas, glaucomatous patients who were diagnosed with glaucoma or a glaucoma follow-up in another health institution before their first presentation, glaucomatous patients diagnosed with acute angle-closure glaucoma, patients who are unable to communicate and patients with an incomplete medical record were excluded from the study.

The sample size was calculated using the double population proportion formula using EPI Info version 7 software. $n = (2 \times P (1-P) (Z_\beta + Z\alpha/2)^2 / (P1—P2)^2$

Where, n = sample size, P = P1+P2; P1 = Proportion of controls with exposure was 19.51%, P2 = Proportion of cases with exposure was 9.67%. Z = the value of z statistic at 95% confidence level = 1.96, $\beta$ power 80% = 0.80, $Z_\beta$ = 0.84, control to case ratio = 1:1 [17], assuming a non-response rate of 10% for cases and controls, the overall sample size was estimated at 492 (246 cases and 246 controls). The study participants for both cases and controls were selected among glaucomatous patients on follow-up who visited the glaucoma clinic during the data collection period using systematic random sampling. The cases were recruited from late glaucoma presenters while the controls were selected among those without late glaucoma. The projected numbers in two months follow-up for cases and controls were 495 and 540 respectively. So, $K_{case}$ = 495/246 = 2.012 approximately 2 and $K_{control}$ = 540/246 = 2.19 approximately 2. Both controls and cases were selected by systemic random sampling with a fraction of k = 2 form their medical record numbers. Then, each selected patient was accessed in the waiting area. If the patient was not available the next immediate medical record number of the same group was selected and a sampling fraction was added to get the next patient. An identification number was given for each medical record number to avoid duplication.

## Operational definitions

**Cases (Late presenters):** Any chronic glaucoma patients diagnosed with cup to disc ratio (CDR) > 0.8, in which there is no suggestion of other optic nerve pathology and typical glaucomatous field loss with a mean deviation of greater than -14 decibel in either of the eyes at their first presentation [16,18].

**Controls:** Chronic glaucoma patients diagnosed with cup to disc ratio (CDR) < 0.8 and typical glaucomatous field loss with a mean deviation of < -14 decibel in both of the eyes at their first presentation [16,18].

**Late glaucoma diagnosis:** Glaucomatous disc cupping of CDR > 0.8, in which there is no suggestion of other optic nerve pathology and typical glaucomatous field loss with a mean deviation of greater than -14 decibel in either of the eyes [18].

**Knowledge:** A standard knowledge questionnaire including seventeen (17) questions was used to assess respondents' knowledge about glaucoma. One point (1) was allocated for each correct response, otherwise, zero (0) was given. Respondents who scored the median ($\geq$7.0) and above of 17 knowledge questions were considered to have good knowledge; while those who scored below the median were considered as having poor knowledge about glaucoma [23].

**Regular eye checkup:** Those individuals who check up their eyes in every two years [17,24].

## Data collection tool and procedure

A semi-structured questionnaire having five parts related to sociodemographic & economic factors, ocular factors, behavioral factors, knowledge-related factors, and systemic disease-related information of the participant was prepared by reviewing different literatures.

A data extraction format was developed to review the chart of each eligible patient to assess the type and stage of glaucoma, IOP, and visual acuity (VA) of study participants. The diagnosis of glaucoma was made by a senior ophthalmologist. The questionnaire was initially prepared in English, translated into Amharic (local language) by language experts for data collection, and re-translated to English to check consistency in meaning of words and concepts. The questionnaire was pre-tested for reliability and validity in 25 glaucomatous patients in another hospital (Bahir Dar Felege Hiwot referral eye Hospital) with the same methods and the content of the questionnaire was assessed for its clarity, completeness and modified accordingly. It was also checked for its reliability using a reliability test and has a Cronbach alpha value of 0.77.

A data collection procedure involving a patient interview and reviewing patients' medical records. The data was collected by trained ophthalmic nurses and supervision was done by a senior optometrist. Data collectors first introduced themselves and the purpose of the study. After obtaining consent from the subjects, data was collected from the participants with face-to face interview. Necessary information was obtained from the patients' medical record that was recorded on their first visit to the glaucoma clinic.

Supervision has been made during the data collection and appropriate feedback had been provided. Training was given to the data collectors before the data collection. Regular check-up for completeness and consistency of the collected data has been made by the principal investigator on daily basis.

## Statistical analysis

Data were coded, entered into EPI Info version 7 (https://www.cdc.gov/epiinfo/pc.html) and exported to SPSS version 22 (https://www.ibm.com/analytics/spss-statistics-softw) for analysis. The descriptive statistics were presented with tables, percentages, mean, and standard deviations. Hosmer- Lemeshow goodness of fit was done to check the model assumption of logistic regression. Multicollinearity between the independent variables was checked using the Variance Inflation Factor and the mean value was less than three. Both bi-variable and multivariable logistic regression analysis was done and variables with p-value < 0.2 under bi-variable logistic regression considered for multivariable logistic regression. In the multivariable logistic regression analysis, variables with a p-value of less than 0.05 were declared as statistically significant. Odds ratio with 95% confidence interval and the corresponding p-value was used to identify determinants of late presentation among glaucoma patients.

## Ethical consideration

Ethical approval was obtained from the Institutional Review Board (IRB) of University of Gondar, College of Medicine and health sciences in accordance with the Declaration of Helsinki. Written letter of permission was obtained from the medical registration office to access the patients' medical record. Written informed consent was obtained from the study participants after a brief explanation of the objective of the study. Any involvement in the study was after their complete consent was obtained. All the study participants were informed about the purpose of the study and their right to refuse and withdraw from the study at any time. Confidentiality was also maintained through an anonymous questionnaire by excluding identifiers and using codes.

## Results

### Socio-demographic characteristics of study participants

From the total of 492 study participants, 452 (226 cases and 226 controls) with a response rate of 91.87% were involved in the study. From the study participants, 277 (61.3%) were males.

The mean age of the study participants at the diagnosis of glaucoma was 55.1 years with a standard deviation (SD) ±13.2 (Table 1).

## Ocular related factors of study participants

The median IOP of the overall study participants was 25.80 mmHg and inter-quartile range (IQR) 9.97 mmHg, while it was 29.00 mmHg and 24.10 mmHg for cases and controls respectively. The most common diagnosis of glaucoma was primary open-angle glaucoma, 255 (56.4%). Of those patients who had IOP > 30 mmHg, 71.53% were cases. Only 71 (15.7%) of the total study participants had habit of regular eye checkup. Among the study participants, 288 (50.44%) had good knowledge about glaucoma (Table 2).

## Determinants of late presentation of glaucoma of study participants

In multivariable logistic regression analysis; age, the distance of residence from UoG TETC, regular eye checkup, high IOP at presentation, knowledge about glaucoma, and history of diabetes mellitus remained significantly associated with late presentation of glaucoma.

**Table 1. Socio-demographic characteristics of study participants in University of Gondar Comprehensive Specialized Hospital, Northwest Ethiopia, 2020.**

| Variable | Controls, n (%) | Cases, n (%) |
|---|---|---|
| Sex | | |
| Male | 142 (62.8) | 135 (59.7) |
| Female | 84 (37.2) | 91(40.3) |
| Age at diagnosis | | |
| 18–40 | 38 (16.8) | 16 (7.1) |
| 41–50 | 72 (31.9) | 48 (21.2) |
| 51–60 | 65 (28.8) | 67 (29.6) |
| >60 | 51 (22.5) | 95 (42.1) |
| Educational status | | |
| No formal education | 81 (35.8) | 102 (45.1) |
| Primary | 31 (13.7) | 37(16.4) |
| Secondary | 54 (23.9) | 52 (23.0) |
| College and above | 60 (26.6) | 35 (15.5) |
| Occupation | | |
| Governmental employee | 38 (16.8) | 20 (8.9) |
| Non-governmental employee | 14 (6.2) | 10 (4.4) |
| Merchant | 67 (29.7) | 65 (28.8) |
| Farmer | 47 (20.8) | 69 (30.5) |
| Housewife | 36 (15.9) | 45 (19.9) |
| Others* | 24 (10.6) | 17 (7.5) |
| Monthly income (US$) | | |
| ≤ 19 | 60 (26.5) | 81 (35.8) |
| 20–33 | 46 (20.4) | 55 (24.4) |
| 34–50 | 39 (17.3) | 38 (16.8) |
| >50 | 81 (35.8) | 52 (23.0) |
| Distance from the hospital in Km | | |
| ≤ 3 | 86 (38.1) | 32 (14.2) |
| 4–24 | 74 (32.7) | 34 (15.1) |
| 25–53 | 35 (15.5) | 81 (35.8) |
| > 53 | 31 (13.7) | 79 (35.0) |
| Positive family history of glaucoma | | |
| Yes | 30 (13.3) | 21 (9.3) |
| No | 173 (76.5) | 182 (80.5) |
| I don't know | 23 (10.2) | 23 (10.2) |

Others* = retired (11), driver (3), daily laborer (9), religious leaders (13), students (5).

**Table 2. Ocular related factors among adult glaucomatous patients in University of Gondar Comprehensive Specialized Hospital, Northwest Ethiopia, 2020.**

| Variable | Controls, n (%) | Cases, n (%) |
|---|---|---|
| Presenting IOP | | |
| <21.00 | 70 (31.0) | 33 (14.6) |
| 21.00–25.00 | 62 (27.4) | 41 (18.1) |
| 25.01–30.00 | 57 (25.2) | 59 (26.1) |
| >30.00 | 37 (16.4) | 93 (41.2) |
| Type of glaucoma | | |
| POAG | 139 (74.3) | 116 (51.8) |
| CACG | 32 (3.1) | 16 (35.4) |
| PxG | 41 (19.9) | 87 (9.7) |
| Others* | 14 (2.7) | 7 (3.1) |
| Systemic diseases | | |
| Diabetes mellitus | | |
| Yes | 59 (26.1) | 15 (6.6) |
| No | 167 (73.9) | 211 (93.4) |
| Hypertension | | |
| Yes | 22 (9.7) | 18 (8.0) |
| No | 204 (90.3) | 208 (92.0) |
| Asthma | | |
| Yes | 7 (3.1) | 11 (4.9) |
| No | 219 (96.9) | 215 (95.1) |
| Previous ocular trauma | | |
| Yes | 18 (8.0) | 8 (3.5) |
| No | 208 (92) | 218 (96.5) |
| Regular eye check up | | |
| Yes | 56 (24.8) | 15 (6.6) |
| No | 170 (75.2) | 211 (93.4) |
| Ocular comorbidity | | |
| No | 165 (73.0) | 159 (70.4) |
| Yes | 61 (27.0) | 67 (29.6) |
| Knowledge about glaucoma | | |
| Poor | 67 (29.6) | 157 (69.5) |
| Good | 159 (70.4) | 69 (30.5) |

Others* = Normal tension glaucoma, Neovascular glaucoma, Steroid induced glaucoma, Phacomorphic glaucoma, POAG = primary open angle glaucoma, CACG = Chronic angle closure glaucoma, PxG = pseudo-exfoliative glaucoma.

Accordingly, being 51–60 and >60 years old had 2.36 times (AOR: 2.36; 95% CI: 1.18, 4.65) and 4.51 times (AOR: 4.51; 95% CI: 1.74, 11.67) higher odds of being late presenter respectively than those ≤ 40 years of age. Participants who resided 24–53 km and > 53 km away from UoG TECTC had the odds of 4.50 times (AOR: 4.50; 2.15, 9.40) and 6.02 times (AOR: 6.02; 2.76, 13.14) more likely being late presenter respectively compared to those who resided <3km away from the UoG TECTC. Similarly, this study revealed that those patients with presenting IOP of 25.01–30.00 mmHg and > 30 mmHg had 2.17 times (AOR: 2.17, 95% CI: 1.23, 5.09) and 4.49 times (AOR: 4.49, 95% CI: 2.10, 9.12) higher odds of presenting late respectively compared to those whose presenting IOP were < 21.00 mmHg. Besides, the odds of late presentation among participants who had poor knowledge of glaucoma was 4.46 times (AOR: 4.46, 95% CI: 2.62, 7.58) higher compared to those who had good knowledge. In the same way, the odds of late presentation for those patients who didn't regularly checkup their eyes was 2.35 times (AOR: 2.35, 95% CI: 1.09, 5.47) more likely to present late compared to those who had regular eye checkup. On the other hand, those who had history of diabetes mellitus had 84% lesser odds of being late presenter (AOR = 0.16, 95% CI: 0.68, 0.38) compared to those who didn't have diabetes (Table 3).

**Table 3. Determinant factors associated with late presentation among adult glaucomatous patients in University of Gondar Comprehensive Specialized Hospital, Northwest Ethiopia, 2020.**

| Study factor | Control | Case | COR (95%CI) | AOR (95% CI) | P value |
|---|---|---|---|---|---|
| Age | | | | | |
| ≤ 40 | 38 | 16 | 1.00 | 1.00 | |
| 41–50 | 72 | 48 | 1.58 (0.79, 3.15) | 1.55 (0.60, 3.13) | 0.086 |
| 51–60 | 65 | 67 | 2.44 (1.20, 4.67) | 2.36 (1.18, 4.65) | **0.025** |
| >60 | 50 | 96 | 4.56 (2.318, 8.97) | 4.51 (1.74, 11.67) | **<0.001** |
| Educational status | | | | | |
| No formal education | 81 | 102 | 2.16 (1.30, 3.60) | 0.52 (0.17,1.56) | 0.241 |
| Primary | 31 | 37 | 2.04 (1.08, 3.85) | 0.66 (0.21, 2.05) | 0.470 |
| Secondary | 54 | 52 | 1.65 (0.94, 2.90) | 1.10 (0.41, 2.93) | 0.852 |
| College & above | 60 | 35 | 1.00 | 1.00 | |
| Occupation | | | | | |
| Governmental | 38 | 20 | 1.00 | 1.00 | |
| Non-governmental | 14 | 10 | 1.36 (0.51, 3.60) | 1.30 (0.36, 4.80) | 0.685 |
| Merchant | 67 | 65 | 1.84 (0.97, 3.50) | 1.46 (0.47, 4.59) | 0.513 |
| Farmer | 47 | 69 | 2.79 (1.46, 5.38) | 0.52 (0.13, 2.12) | 0.367 |
| House wife | 36 | 45 | 2.38 (1.18, 4.77) | 0.74 (0.19, 2.94) | 0.666 |
| Others | 24 | 17 | 1.35 (0.59, 3.07) | 1.82 (0.45, 7.30) | 0.400 |
| Monthly income (US$) | | | | | |
| ≤ 19 | 60 | 81 | 2.10 (1.30, 3.40) | 2.29 (0.86, 5.90) | 0.106 |
| 20–33 | 46 | 55 | 1.87 (1.10, 3.14) | 1.30 (0.50, 3.39) | 0.591 |
| 34–50 | 39 | 38 | 1.52 (0.86, 2.67) | 1.13 (0.491, 2.60) | 0.774 |
| > 50 | 81 | 52 | 1.00 | 1.00 | |
| Distance in Km | | | | | |
| ≤3 km | 86 | 32 | 1.00 | 1.00 | |
| 4–24 km | 74 | 34 | 1.23 (0.69, 2.19) | 1.06 (0.52, 2.18) | 0.871 |
| 25–53 | 35 | 81 | 6.22 (3.52, 10.96) | 4.50 (2.15, 9.40) | **<0.001** |
| > 53 | 31 | 79 | 6.85 (3.83, 12.24) | 6.02 (2.76, 13.14) | **<0.001** |
| Regular eye checkup | | | | | |
| Yes | 56 | 15 | 1.00 | 1.00 | |
| No | 170 | 211 | 4.63 (2.53,8.48) | 2.35 (1.09, 5.47) | **0.044** |
| Diabetes mellitus | | | | | |
| Yes | 59 | 15 | 0.20 (0.11, 0.37) | 0.16 (0.68, 0.38) | **<0.001** |
| No | 167 | 211 | 1.00 | 1.00 | |
| Ocular injury | | | | | |
| Yes | 18 | 8 | 0.42 (0.18, 0.99) | 0.45 (0.15, 1.32) | 0.146 |
| No | 208 | 218 | 1.00 | 1.00 | |
| Knowledge about glaucoma | | | | | |
| Poor | 67 | 157 | 5.40 (3.61, 8.07) | 4.46 (2.62, 7.58) | **<0.001** |
| Good | 159 | 69 | 1.00 | 1.00 | |
| Pseudo-exfoliation | | | | | |
| No | 185 | 132 | 1.00 | 1.00 | |
| Yes | 41 | 94 | 3.21 (2.10, 4.93) | 0.36 (0.12, 1.07) | 0.066 |
| IOP | | | | | |
| <21.00 | 70 | 33 | 1.00 | 1.00 | |
| 21.00–25.00 | 62 | 41 | 1.40 (0.80, 2.48) | 1.33 (0.77, 3.86) | 0.111 |
| 25.01–30.00 | 57 | 59 | 2.19 (1.26, 3.81) | 2.17 (1.23, 5.09) | **0.011** |
| >30.00 | 37 | 93 | 5.33 (3.04, 9.36) | 4.49 (2.10, 9.12) | **<0.001** |
| Type of glaucoma | | | | | |
| POAG | 139 | 116 | 1.00 | 1.00 | |
| CACG | 32 | 16 | 0.60 (0.31, 1.15) | 0.50 (0.21, 1.18) | 0.113 |
| PxG | 41 | 87 | 2.54 (1.63, 3.97) | 0.71 (0.23, 2.14) | 0.543 |
| Others | 14 | 7 | 0.60 (0.23, 1.53) | 0.65 (0.21, 1.97) | 0.440 |

COR = Crudes Odds Ratio, AOR = Adjusted Odds Ratio, CI = Confidence Interval, **Bolded figures** = statistically significant, POAG = Primary Open Angle Glaucoma, CACG = Chronic Angle ClosureGglaucoma, PxG = Pseudo-exfoliative Glaucoma.

## Discussion

This study attempts to elucidate determinants for late presentation of glaucoma among glaucomatous patients.

In this study, age was an independent factor for the late presentation of glaucoma. A strong positive relationship between increasing age and risk of the late presentation was seen. By which those > 60 years of age were 4.51 times more likely to present late. Similar results were reported from different studies [17,19,20–21,25,26]. This might be because the prevalence and incidence of glaucoma increase with age [4] and it might also be explained by the low health-care-seeking behavior of elderly individuals [27].

This study also revealed high IOP at presentation as a determinant for late presentation of glaucoma. By which those who presented with IOP between 25.01–30.00 mmHg and >30.00 mmHg became 2.17 times and 4.49 times more likely to present late respectively than those who presented with IOP $\leq$ 21.00mmHg. The result is in line with other studies which showed higher IOPs result in more rapid visual field damage and increased risk of late presentation [16,18,19]. This might be due to the evidence that higher IOPs lead to more rapid visual field loss [28].

The distance of residency from the hospital was also found to be significantly associated with the late presentation of patients. This might be due to the reason that the geographic proximity of the health care center has a substantial impact on the health-seeking behavior of patients [29].

Another significant association with the late presentation of glaucoma in this study was poor knowledge about glaucoma. Those individuals with poor knowledge about the disease were late presenters compared to those with good knowledge about the disease. Similar findings were also reported from other studies [21,26]. This could be due to the reason that having good knowledge about glaucoma as a blinding and irreversible disease influences the eye care service-seeking behavior of people and their uptake of services [30]. However, a study done in South Africa [18] revealed no significant association between knowledge of glaucoma and late presentation. In the South African study, knowledge of glaucoma as a blinding disease was assessed using only a single question, which is not standardized. While the present study used seventeen standard questions which were relatively more detailed to assess every dimension of patients' knowledge on glaucoma, which might explain the discrepancy. Moreover, the smaller sample size (66 cases and 66 controls) recruited in the South African study might mask the association.

This study sought a significant association between regular eye check-ups and late presentation of glaucoma. This is comparable with another study done in the United Kingdom and Iran [16,17]. This might be since those who attend regular eye check-ups are more likely to seek medical attention earlier in their eye disease. These results lend weight to the concept that those who did not regularly check their sight tests are at greater risk of late presentation.

The present study also revealed that patients who had a history of diabetes mellitus are less likely to present late compared to those who didn't have diabetes mellitus. The result is comparable with a study done in South Africa [18]. This might be due to the reason that; diabetic patients are more likely to have regular medical and ocular examinations for diabetic retinopathy screening and follow-up, hence the opportunity to spot glaucoma at an earlier stage. This can be supported by the evidence of opportunistic detection of glaucomatous optic discs within a diabetic retinopathy screening [31].

This study might have inherited limitation of recall bias due to the study design. When knowledge of participants on glaucoma was assessed, it was their current knowledge that was assessed, and this might have an impact on the participant's knowledge report. The history of diabetes mellitus was self-reported by the study participants.

## Conclusion

Increasing age and high IOP have substantial positive association with late presentation of glaucoma. Moreover, having poor knowledge about glaucoma, absence of regular eye check-up and being resided at far distance were positively associated with late presentation. Nevertheless, having history of diabetes mellitus was associated negatively with late presentation of glaucoma.

## Supporting information

**S1 Questionnaire. Questionnaire in English Language.**
(PDF)

**S2 Questionnaire. Questionnaire in Amharic language.**
(PDF)

**S1 Data. Sample data for determinants and late presentation of glaucoma.**
(XLSX)

## Acknowledgments

We are deeply indebted the University of Gondar for gave us ethical clearance to conduct this research. We would also like to acknowledge study participants for their cooperation and willingness during the data collection.

## Author Contributions

**Conceptualization:** Biruktayit Kefyalew Belete, Natnael Lakachew Assefa, Abel Sinshaw Assem, Fisseha Admasu Ayele.

**Data curation:** Biruktayit Kefyalew Belete, Natnael Lakachew Assefa, Abel Sinshaw Assem, Fisseha Admasu Ayele.

**Formal analysis:** Biruktayit Kefyalew Belete, Natnael Lakachew Assefa, Abel Sinshaw Assem, Fisseha Admasu Ayele.

**Funding acquisition:** Biruktayit Kefyalew Belete, Natnael Lakachew Assefa, Abel Sinshaw Assem, Fisseha Admasu Ayele.

**Investigation:** Biruktayit Kefyalew Belete, Natnael Lakachew Assefa, Abel Sinshaw Assem, Fisseha Admasu Ayele.

**Methodology:** Biruktayit Kefyalew Belete, Natnael Lakachew Assefa, Abel Sinshaw Assem, Fisseha Admasu Ayele.

**Project administration:** Biruktayit Kefyalew Belete, Natnael Lakachew Assefa, Abel Sinshaw Assem, Fisseha Admasu Ayele.

**Resources:** Biruktayit Kefyalew Belete, Natnael Lakachew Assefa, Abel Sinshaw Assem, Fisseha Admasu Ayele.

**Software:** Biruktayit Kefyalew Belete, Natnael Lakachew Assefa, Abel Sinshaw Assem, Fisseha Admasu Ayele.

**Supervision:** Biruktayit Kefyalew Belete, Natnael Lakachew Assefa, Abel Sinshaw Assem, Fisseha Admasu Ayele.

**Validation:** Biruktayit Kefyalew Belete, Natnael Lakachew Assefa, Abel Sinshaw Assem, Fisseha Admasu Ayele.

**Visualization:** Biruktayit Kefyalew Belete, Natnael Lakachew Assefa, Abel Sinshaw Assem, Fisseha Admasu Ayele.

**Writing – original draft:** Biruktayit Kefyalew Belete, Natnael Lakachew Assefa, Abel Sinshaw Assem, Fisseha Admasu Ayele.

**Writing – review & editing:** Biruktayit Kefyalew Belete, Natnael Lakachew Assefa, Abel Sinshaw Assem, Fisseha Admasu Ayele.

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
