## [Decision Letter · Decision Letter 0]

17 Nov 2021

PONE-D-21-27844Determinants for late presentation of glaucoma among adult glaucomatous patients in University of Gondar Comprehensive Specialized Hospital: Case-control studyPLOS ONE

Dear Dr. Assefa,

Thank you for submitting your manuscript to PLOS ONE. After careful consideration, we feel that it has merit but does not fully meet PLOS ONE’s publication criteria as it currently stands. Therefore, we invite you to submit a revised version of the manuscript that addresses the points raised during the review process.

We look forward to receiving your revised manuscript.

Kind regards,

David Chau

Academic Editor

PLOS ONE

Journal Requirements:

3. Please include additional information regarding the survey or questionnaire used in the study and ensure that you have provided sufficient details that others could replicate the analyses. For instance, if you developed a questionnaire as part of this study and it is not under a copyright more restrictive than CC-BY, please include a copy, in both the original language and English, as Supporting Information

4. Please provide additional details regarding participant consent. In the ethics statement in the Methods and online submission information, please ensure that you have specified whether consent was written or verbal/oral. If consent was verbal/oral, please specify: 1) whether the ethics committee approved the verbal/oral consent procedure, 2) why written consent could not be obtained, and 3) how verbal/oral consent was recorded. If your study included minors, please state whether you obtained consent from parents or guardians in these cases. If the need for consent was waived by the ethics committee, please include this information

Reviewers' comments:

Reviewer's Responses to Questions

**Comments to the Author**

1. Is the manuscript technically sound, and do the data support the conclusions?

Reviewer #1: Yes

Reviewer #2: Partly

Reviewer #3: Yes

2. Has the statistical analysis been performed appropriately and rigorously? 

Reviewer #1: Yes

Reviewer #2: N/A

Reviewer #3: Yes

3. Have the authors made all data underlying the findings in their manuscript fully available?

Reviewer #1: Yes

Reviewer #2: Yes

Reviewer #3: No

4. Is the manuscript presented in an intelligible fashion and written in standard English?

Reviewer #1: No

Reviewer #2: No

Reviewer #3: Yes

5. Review Comments to the Author

Reviewer #1: The study attempted a very important subject matter relevant to glaucoma management in an African population. The authors however need to address the following to make the study publishable.

1. The results section of the abstract is not precise and they needs to be refocused.

2. The glaucoma diagnosis criteria should be clearly stated to provide context for the glaucoma diagnosis to provide a basis for comparison with other studies.

3. The classification of age is arbitrary i.e 18-40 then 41-50. Please use standard age classification such as 18-35 youth, 36-59 adult, 60 and older.

4. Was only eye involved in the analysis where the glaucoma was bilateral? was there a correlation between the 2 eyes? if so then stick the use of one eye in the analysis.

5. How was Normal Tension Glaucoma diagnosed? Please explicitly outline.

6. The table headings are to laborious. Authors should relook at them

Reviewer #2: The authors attempted to find determinants for late presentation of glaucoma

among adult glaucomatous patients. While the effort is commendable, the

rationale/ basis is not strong, and the value-add of this work seems minimally

incremental. In general, more details need to be added in methods section.

I have several minor and major comments.

Major comments：

1．The experimental design is not rigorous and solid enough, also the results are not strong enough to support conclusion.

2．Page 6. Methods: the definition of late presentation of glaucoma and controls are not clear, please add more detailed information about the selection of cases and controls (Please reference to Fraser S, Bunce C, Wormald R, Brunner E. Deprivation and late presentation of glaucoma: case-control study. BMJ. 2001 Mar 17;322(7287):639-43. doi: 10.1136/bmj.322.7287.639. PMID: 11250847; PMCID: PMC26542; Motlagh BF, Pirbazari TJ. Risk factors for late presentation of chronic glaucoma in an Iranian population. Oman J Ophthalmol. 2016 May-Aug;9(2):97-100. doi: 10.4103/0974-620X.184527. PMID: 27433036; PMCID: PMC4932803.).

3．Page 10. Table 1. The classification of patients’ occupation is not logical enough. The standard occupational classification categorized into three groups: professional and technical occupations (I and II); manual and nonmanual skilled occupations (III and IV) and partly skilled and nonskilled occupations (V and VI). Please explain the reason about the classification in your manuscript.

Minor comments:

1．Page 3, introduction: “In 2013, the number of people aged 40 to 80 years with glaucoma was 64.3 million which was predicted to increase to 76.0 million in 2020 and 111.8 million in 2040”. Reference should be added.

2．Page 3, introduction: “In Africa and South Asia, the prevalence of undiagnosed glaucoma in the population has been reported to be more than 90%”. Please modify the sentence, current one is hard to understand.

3．Page 4, introduction: More information should be added to describe the significant of this study?

4．Page 6: Methods: typo, it should be “systematic random sampling” but not “systemic random sampling”.

5．Page 6: Methods: Please provide more information about the questioner of knowledge-related factors in supplementary material.

6．Page 7: Methods: Definition about “Regular eye checkup”, dose it means eye checkup for every two years or only once in past two years? If so, it should not be defined as “regular” eye check-up but “frequency of ophthalmic examination in the last 2 years”.

7．Page 8: Methods: Software information (EPI Info, SPSS) should be provided, for example, the company name or website link.

8．Page 10. Results, table 1. Are there any patients under 18 years old? And please clarify the unit for monthly income?

9．Page 15. Results, table 3. Monthly income in table 3 is not consistent with table 1. Please keep it consistent in the whole manuscript.

10．Page 16. Results, table 3. Please define " POAG, CACG, PxG" in table 3 legend.

Generally, the manuscript is well-written and coherent but there are some minor issues in grammar, spelling and usage that could be remedied by closer copy-editing. Please consider English editing as regards using singular versus plural in verbs, kindly go through the whole manuscript to review this linguistic issue.

Reviewer #3: The manuscript is very well written. The study used a case-control study design. Such design type requires a careful selection of controls. The design and control selection was appropriate and was described clearly. The data analysis and results presentation and interpretation were all reasonable and easy for readers to follow.

A main suggestion is to interpret the results in the public health context, i.e. to discuss what can be done from a public health perspective on timely glaucoma diagnosis based the observed results. The current Conclusion basically repeats the data results after a literature review, but what is really more important (and why the study was done in the first place) is how the study results and knowledge can inform potential interventions to prevent late glaucoma diagnosis in a typic resource limited setting in Africa, e.g. community based educational campaigns, information brochure in diabetes clinics etc.

Some minor comments:

Page 3, "In Africa and South Asia, the prevalence of undiagnosed glaucoma in the population has been reported to be more than 90%", this seems unbelievably high: >90% in the population have glaucoma. Glaucoma prevalence can't be this high.

Page 11, the middle, "The median IOP of the over study participants was 25.80 mmHg and ...". Is this reporting the IOP at diagnosis time, or the IOP at time of this study's visit?

6. PLOS authors have the option to publish the peer review history of their article (what does this mean?). If published, this will include your full peer review and any attached files.

Reviewer #1: No

Reviewer #2: No

Reviewer #3: No

---

## [Author Response · Author response to Decision Letter 0]

2 Feb 2022

Authors’ response for the Editorials and reviewers’ comment 

Manuscript number: PONE-D-21-27844

Manuscript title: Determinants for late presentation of glaucoma among adult glaucomatous patients in University of Gondar Comprehensive Specialized Hospital: Case-control study

Responses to the Editorials 

Authors’ Response: Thank you for your comment! 

• We accepted the comment, 

• We checked and corrected based on the journal requirements point-by-point.

2. We suggest you thoroughly copyedit your manuscript for language usage, spelling, and grammar. If you do not know anyone who can help you do this, you may wish to consider employing a professional scientific editing service. Upon resubmission, please provide the following: The name of the colleague or the details of the professional service that edited your manuscript. A copy of your manuscript showing your changes by either highlighting them or using track changes (uploaded as a *supporting information* file) A clean copy of the edited manuscript (uploaded as the new *manuscript* file)

Authors’ Response: Thank you for your suggestions! 

• We accepted the comment,

• All files were uploaded based on the requirement accordingly. 

• All of the authors are participated and reviewed the manuscript for the language usage, spelling, and grammar.

• Specially, the authors listed below (The higher institutional employs who had “English language proficiency”) reviewed the manuscript in-depth.

• Mr. Natnael Lakachew Assefa (Assistant Professor of Optometry): Reviewed the document, responded the editors and reviewers comment, checked the language usage, spelling error, and grammar correction.

• Mr. Abel Sinshaw Assem (Lecturer of Clinical Optometry): Reviewed the manuscript, checked the language usage, spelling error, and grammar correction.

3. Please include additional information regarding the survey or questionnaire used in the study and ensure that you have provided sufficient details that others could replicate the analyses. For instance, if you developed a questionnaire as part of this study and it is not under a copyright more restrictive than CC-BY, please include a copy, in both the original language and English, as Supporting Information

Authors’ Response: Thank you for your information! 

• We accepted the comment and submitted the data collection tool (Questionnaire) as supporting Information.

• “S1 Questionnaire in English Language” and “S2 Questionnaire in Amharic (Local Language)”.

4. Please provide additional details regarding participant consent. In the ethics statement in the Methods and online submission information, please ensure that you have specified whether consent was written or verbal/oral. If consent was verbal/oral, please specify: 1) whether the ethics committee approved the verbal/oral consent procedure, 2) why written consent could not be obtained, and 3) how verbal/oral consent was recorded. If your study included minors, please state whether you obtained consent from parents or guardians in these cases. If the need for consent was waived by the ethics committee, please include this information

Authors’ Response: Thank you for your comment! 

• We accepted the comment and described in detail

• “Ethical requirement: Page 9, line 2-10” and “Consent form: S1, Page 1”

5. PLOS requires an ORCID iD for the corresponding author in Editorial Manager on papers submitted after December 6th, 2016. Please ensure that you have an ORCID iD and that it is validated in Editorial Manager. 

Authors’ Response: Thank you for your comment! 

• We accepted the comment and authorization accessed with this link https://orcid.org/0000-0002-4998-6128

6. In your Data Availability statement, you have not specified where the minimal data set underlying the results described in your manuscript can be found. PLOS defines a study's minimal data set as the underlying data used to reach the conclusions drawn in the manuscript and any additional data required to replicate the reported study findings in their entirety. Authors’ Response: Thank you for your comment! 

• We accepted the comment and uploaded sample data with this submission as a supporting Information: “S3- Sample data” 

Responses to the Reviewer #1

1. The results section of the abstract is not precise and they needs to be refocused.

Authors’ Response: Thank you for your comment! 

• We accepted the comment and revised the abstract section including the result section.

• “Abstract result section: Page 2, line 13-17”

2. The glaucoma diagnosis criteria should be clearly stated to provide context for the glaucoma diagnosis to provide a basis for comparison with other studies.

Authors’ Response: Thank you for your comment! 

• We accepted the comment and corrected.

• “Methods section: page 6, line 18-20”: “Late glaucoma diagnosis: Glaucomatous disc cupping of CDR > 0.8, in which there is no suggestion of other optic nerve pathology and typical glaucomatous field loss with a mean deviation of greater than -14 decibel in either of the eyes.”

3. The classification of age is arbitrary i.e 18-40 then 41-50. Please use standard age classification such as 18-35 youth, 36-59 adult, 60 and older.

Authors’ Response: Thank you for your comment! 

• Age classification was done by SPSS software analysis based on the participant’s age range with interquartile method. 

• The study participants included in our study were aged ≥ 18 years.

• Due to study subjects age variation and limited studies with same population characteristics in the study area, we classified based on statistics. 

• If we used the suggested classification, most of the subjects would be in the same age group and difficult for analysis. 

4. Was only eye involved in the analysis where the glaucoma was bilateral? was there a correlation between the 2 eyes? if so then stick the use of one eye in the analysis.

Authors’ Response: Thank you for your comment! 

• All patients who fulfill the Late glaucoma diagnosis criteria of CDR > 0.8 and MD > -14 decibel in either eye (Either both eyes or one eye) were included in the analysis. 

• “Described in Methods section: page 6, line 11-17”:

• Cases (Late presenters): Any chronic glaucoma patients diagnosed with cup to disc ratio (CDR) > 0.8, in which there is no suggestion of other optic nerve pathology and typical glaucomatous field loss with a mean deviation of greater than -14 decibel in either of the eyes at their first presentation [16, 18].

• Controls: Chronic glaucoma patients diagnosed with cup to disc ratio (CDR) < 0.8 and typical glaucomatous field loss with a mean deviation of < -14 decibel in both of the eyes at their first presentation [16, 18].

5. How was Normal Tension Glaucoma diagnosed? Please explicitly outline

Authors’ Response: Thank you for your comment! 

• Since the diagnosis of all chronic glaucoma types were done by CDR and MD, Normal Tension Glaucoma was diagnosed with same criteria.

• “Described in Methods section: page 6, line 18-20”: “Late glaucoma diagnosis: Glaucomatous disc cupping of CDR > 0.8, in which there is no suggestion of other optic nerve pathology and typical glaucomatous field loss with a mean deviation of greater than -14 decibel in either of the eyes.”

6. The table headings are to laborious. Authors should relook at them:

Authors’ Response: Thank you for your comment! 

• We accepted the comment and revised table headings:

• “Result section: page 10, line 7-8; page 12, line 9-10; page 15, line 4-5”

Responses to the Reviewer #2

Major comments:

1. The experimental design is not rigorous and solid enough, also the results are not strong enough to support conclusion

Authors’ Response: Thank you for your comment! 

• We accepted the comment and we appreciated your suggestion: 

• But the study design employed was cross-sectional not experimental: “Methods section: page 5, line 3”.

• This study might have an inheritance limitation of the cross-sectional study design for the conclusion of cause and effects due to this we included this as a limitation of our study at the end of the discussion section: “Discussion section: page 20, line 5-8”.

2. Page 6. Methods: the definition of late presentation of glaucoma and controls are not clear, please add more detailed information about the selection of cases and controls (Please reference to Fraser S, Bunce C, Wormald R, Brunner E. Deprivation and late presentation of glaucoma: case-control study. BMJ. 2001 Mar 17;322(7287):639-43. doi: 10.1136/bmj.322.7287.639. PMID: 11250847; PMCID: PMC26542; Motlagh BF, Pirbazari TJ. Risk factors for late presentation of chronic glaucoma in an Iranian population. Oman J Ophthalmol. 2016 May-Aug;9(2):97-100. doi: 10.4103/0974-620X.184527. PMID: 27433036; PMCID: PMC4932803.).

Authors’ Response: Thank you for your comment! 

• We accepted the comment and revised the definitions of cases and controls based on your suggestion in the corrected manuscript operational definition section. 

• “Described in Methods section: page 6, line 11-17”:

• Cases (Late presenters): Any chronic glaucoma patients diagnosed with cup to disc ratio (CDR) > 0.8, in which there is no suggestion of other optic nerve pathology and typical glaucomatous field loss with a mean deviation of greater than -14 decibel in either of the eyes at their first presentation [16, 18].

• Controls: Chronic glaucoma patients diagnosed with cup to disc ratio (CDR) < 0.8 and typical glaucomatous field loss with a mean deviation of < -14 decibel in both of the eyes at their first presentation [16, 18].

3. Page 10. Table 1. The classification of patients’ occupation is not logical enough. The standard occupational classification categorized into three groups: professional and technical occupations (I and II); manual and nonmanual skilled occupations (III and IV) and partly skilled and nonskilled occupations (V and VI). Please explain the reason about the classification in your manuscript.

Authors’ Response: Thank you for your comment! 

• We classified occupations based on the study participants’ occupational statues and considering the general population occupation classification of the study setting/area. 

• Those suggested classification is not familiar classification in our settings.

• The study population occupations are categorized in one of the listed occupations in our study area, 

• That was the reason and used these classifications to be consistent with other studies. 

Miner comments:

1. Page 3, introduction: “In 2013, the number of people aged 40 to 80 years with glaucoma was 64.3 million which was predicted to increase to 76.0 million in 2020 and 111.8 million in 2040”. Reference should be added.

Authors’ Response: Thank you for your comment! 

• We accepted the comment and reference is added in the corrected manuscript.

• “Tham YC, Li X, Wong TY, Quigley HA, Aung T, Cheng CY. Global prevalence of glaucoma and projections of glaucoma burden through 2040: a systematic review and meta-analysis. Ophthalmology 2014, 121(11):2081-2090.”

2. Page 3, introduction: “In Africa and South Asia, the prevalence of undiagnosed glaucoma in the population has been reported to be more than 90%”. Please modify the sentence, current one is hard to understand.

Authors’ Response: Thank you for your comment! 

• We accepted the comment and rewritten the sentence clearly in the corrected manuscript. 

• “Introduction section: page 3, line 7-10”: “Population-based studies in Asian countries showed a higher prevalence of glaucoma [3] and Primary Open Angle Glaucoma (POAG) is the most commonly reported [4-6]. The prevalence of previously undiagnosed glaucoma in South Africa was 87.0% [7].”

3. Page 4, introduction: More information should be added to describe the significant of this study?

Authors’ Response: Thank you for your comment! 

• We accepted the comment and included more descriptive sentences for the significance of the study in the corrected manuscript. 

• “Introduction section: page 4, line 1-3 and line 7-9”

4. Page 6: Methods: typo, it should be “systematic random sampling” but not “systemic random sampling”.

Authors’ Response: Thank you for your comment! 

• It was systematic random sampling method and described in the corrected manuscript methods section: 

• “Methods section: page 5, line 21-23 and page 6, line 1-6”: “The study participants for both cases and controls were selected among glaucomatous patients on follow-up who visited the glaucoma clinic during the data collection period using systematic random sampling. The cases were recruited from late glaucoma presenters while the controls were selected among those without late glaucoma. The projected numbers in two months follow-up for cases and controls were 495 and 540 respectively. So, Kcase = 495/246= 2.012 approximately 2 and Kcontrol = 540/246 = 2.19 approximately 2. Both controls and cases were selected by systemic random sampling with a fraction of k=2 form their medical record numbers.”

5. Page 6: Methods: Please provide more information about the questioner of knowledge-related factors in supplementary material.

Authors’ Response: Thank you for your comment! 

• We accepted the comment and provided the questionnaire as supplementary document in English language and Amharic (Local language|). 

• “S1 Questionnaire in English Language and S2 Questionnaire in Amharic (Local Language)”

6. Page 7: Methods: Definition about “Regular eye checkup”, dose it means eye checkup for every two years or only once in past two years? If so, it should not be defined as “regular” eye check-up but “frequency of ophthalmic examination in the last 2 years”.

Authors’ Response: Thank you for your comment! 

E We accepted the comment and corrected in the revised manuscript as “Those individuals who check up their eyes every two years”.

E “Methods section: page 7, line 4-5”: “Regular eye checkup: Those individuals who check up their eyes in every two years [17, 24].”

7. Page 8: Methods: Software information (EPI Info, SPSS) should be provided, for example, the company name or website link.

Authors’ Response: Thank you for your comment! 

• We accepted the comment and indicted the company name in the revised manuscript.

• “Methods section: page 8, line 9-10” :

• “EPI INFO Version 7: https://www.cdc.gov/epiinfo”

• “SPSS Version 20: https://www.ibm.com/analytics/spss-statistics-software”

8. Page 10. Results, table 1. Are there any patients under 18 years old? And please clarify the unit for monthly income?

Authors’ Response: Thank you for your comment! 

• No: The study participants included in the study were adults aged equally or greater than 18 years which is described in the revised manuscript methods section.

• “Methods section: page 5, line 9-10”: All adult glaucomatous patients aged ≥, diagnosed within the last two years and on follow-up were included in the study.

• Unit monthly income was “US dollars (US$)” and corrected in the revised manuscript. “Result section: page 11, table 1 and page 16, table 2” 

9. Page 15. Results, table 3. Monthly income in table 3 is not consistent with table1. Please keep it consistent in the whole manuscript.

Authors’ Response: Thank you for your comment! 

• We accepted the comment and corrected in the revised manuscript.

10. Page 16. Results, table 3. Please define " POAG, CACG, PxG" in table 3 legend.

Authors’ Response: Thank you for your comment! 

• We accepted the comment and corrected in the revised manuscript.

Responses to the Reviewer #3

1. A main suggestion is to interpret the results in the public health context, i.e. to discuss what can be done from a public health perspective on timely glaucoma diagnosis based the observed results. The current Conclusion basically repeats the data results after a literature review, but what is really more important (and why the study was done in the first place) is how the study results and knowledge can inform potential interventions to prevent late glaucoma diagnosis in a typic resource limited setting in Africa, e.g. community based educational campaigns, information brochure in diabetes clinics etc.

Authors’ Response: Thank you for your comment! 

• We accepted the comment and suggestions: 

• We added the significant of the study in the introduction section, conclusions were revised based on the objective of the study: “Introduction section: page 4, line 1-3 and line 7-9”

• We revised the introduction section (the state of knowledge, significance of the study for the population and others): “Introduction section: page 3, line 21-23 and page 4, line 1-6”.

• We revised the discussions and conclusions based on your comment and suggestions.

2. Page 3, "In Africa and South Asia, the prevalence of undiagnosed glaucoma in the population has been reported to be more than 90%", this seems unbelievably high: >90% in the population have glaucoma. Glaucoma prevalence can't be this high.

Authors’ Response: Thank you for your comment! 

• We accepted the comment and revised the references and modified accordingly.

• “Introduction section: page 3, line 7-10”: “Population-based studies in Asian countries showed a higher prevalence of glaucoma [3] and Primary Open Angle Glaucoma (POAG) is the most commonly reported [4-6]. The prevalence of previously undiagnosed glaucoma in South Africa was 87.0% [7].”

3. Page 11, the middle, "The median IOP of the over study participants was 25.80 mmHg and ...". Is this reporting the IOP at diagnosis time, or the IOP at time of this study's visit?

Authors’ Response: Thank you for your comment! 

• The data for IOP was taken from the patients’ medical record card at thier first diagnosis time not during the data collection time of the study. 

• Described in the manuscript methods section: page 7, line 22-23 and in the uploaded supporting information S1 and S2 data collection tool.

---

## [Decision Letter · Decision Letter 1]

24 Feb 2022

PONE-D-21-27844R1Determinants for late presentation of glaucoma among adult glaucomatous patients in University of Gondar Comprehensive Specialized Hospital. Case-control studyPLOS ONE

Dear Dr. Assefa,

Thank you for submitting your manuscript to PLOS ONE. After careful consideration, we feel that it has merit but does not fully meet PLOS ONE’s publication criteria as it currently stands. Therefore, we invite you to submit a revised version of the manuscript that addresses the points raised during the review process.

We look forward to receiving your revised manuscript.

Kind regards,

David Chau

Academic Editor

PLOS ONE

Journal Requirements:

Reviewers' comments:

Reviewer's Responses to Questions

**Comments to the Author**

1. If the authors have adequately addressed your comments raised in a previous round of review and you feel that this manuscript is now acceptable for publication, you may indicate that here to bypass the “Comments to the Author” section, enter your conflict of interest statement in the “Confidential to Editor” section, and submit your "Accept" recommendation.

Reviewer #1: All comments have been addressed

Reviewer #4: (No Response)

2. Is the manuscript technically sound, and do the data support the conclusions?

Reviewer #1: Yes

Reviewer #4: Partly

3. Has the statistical analysis been performed appropriately and rigorously? 

Reviewer #1: Yes

Reviewer #4: Yes

4. Have the authors made all data underlying the findings in their manuscript fully available?

Reviewer #1: Yes

Reviewer #4: Yes

5. Is the manuscript presented in an intelligible fashion and written in standard English?

Reviewer #1: Yes

Reviewer #4: No

6. Review Comments to the Author

Reviewer #1: The correction indicated by the reviewer have been addressed and the now the paper reads more coherently and structurally conforms to the format of the journal.

Reviewer #4: This is an interesting study, although could be improved by a clearer argument for why previous literature from the area is limited and/or inappropriate, and so why this specific study is required. There are some grammatical slips throughout the manuscript, although these are minor. Specific comments are outlined, below.

Abstract

- two decimal places for age is probably excessive (similarly elsewhere in the manuscript)

- the abstract doesn’t give a definition of “late presenter”

- “residing far away from the hospital” mention in the Conclusion is not listed in the results

Introduction

- Pg 3, line 14: grammar

- The introduction could make a more clearly argued case for why the study is important: in particular, it cites three studies regarding what are the determinants of late presentation (including one from Africa), but then goes on to say “there is little published evidences [sic] for why some… have advanced disease at first diagnosis”. What are the gaps and/or conflicts in previous work that into this area, and how does the current study hope to address these? in particular, if the authors feel that the previous work from elsewhere in Africa is not applicable to Ethiopia, they should probably state so more explicitly: even better if they can cite supporting demographic / other evidence to support why this is the case. Some of the arguments made in the Discussion regarding weaknesses in previous studies from Africa (e.g. regarding sample size) could be mentioned explicitly in the Introduction.

Materials and Methods

- pg 5, Line 9: age criterion is missing

- pg 6, line 21: questionnaire

- pg 7, line 15: “expertise”: experts?

- pg 7, line 16: a bit more detail about the reliability check would be useful (assuming it involved retesting, how many people were retested, and what was the interval between testing?): or is this the information that is partly provided on page 8, line 1?

Results

It is stated that a history of diabetes was significantly associated with late presentation: isn’t it the ABSENCE of a history of diabetes that is associated?

Discussion

pg 18, line 20: grammar (sentence fragment)

pg 18, line 23: these cited reasons are rather generic: a much better job of targeting specific reasons is done in the following paragraph when differences between the current study, and the previous South African study, are discussed

7. PLOS authors have the option to publish the peer review history of their article (what does this mean?). If published, this will include your full peer review and any attached files.

Reviewer #1: **Yes: **Samuel Kyei

Reviewer #4: No

---

## [Author Response · Author response to Decision Letter 1]

1 Mar 2022

Authors’ response for the Editorials and reviewers’ comment 

Manuscript number: PONE-D-21-27844

Manuscript title: Determinants for late presentation of glaucoma among adult glaucomatous patients in University of Gondar Comprehensive Specialized Hospital: Case-control study

Responses to the Editorials 

1. Journal Requirements:

Authors’ Response: Thank you for your comment! 

• We checked each references and corrected based on the journal reference guidelines.

• We reviewed and checked the document point by point based on the comments and suggestions of the editor’s and reviewers’.

Responses to the Reviewer #1

1. Reviewer #1: The correction indicated by the reviewer have been addressed and the now the paper reads more coherently and structurally conforms to the format of the journal.

Authors’ Response: Thank you in advance for your constructive comment and sharing your profound experience throughout the manuscript revision process. 

Responses to the Reviewer #4

1. Reviewer #4: This is an interesting study, although could be improved by a clearer argument for why previous literature from the area is limited and/or inappropriate, and so why this specific study is required. There are some grammatical slips throughout the manuscript, although these are minor. Specific comments are outlined, below.

Authors’ Response: Thank you in advance for your constructive comment and suggestions. 

Abstract

1. two decimal places for age is probably excessive (similarly elsewhere in the manuscript)

Authors’ Response: we accepted and corrected with one decimal places in the new manuscript.

2. the abstract doesn’t give a definition of “late presenter”

Authors’ Response: we accepted and definition included in the new manuscript: 

• Page 2, line 7-9: “Late presenters were glaucoma patients diagnosed with cup to disc ratio (CDR) > 0.8 and mean deviation of greater than -14 decibel in either of the eyes at their first presentation.”

3. “residing far away from the hospital” mention in the Conclusion is not listed in the results

Authors’ Response: These are resided > 53 km (the farthest km) away from the hospital which was described in: 

• Abstract results section: page 2, line 15: “resided > 53 km away from the hospital 6.02 times (AOR: 6.02; 2.76, 13.14”

• Results section: page 14, line 10-13: “Participants who resided 24 – 53 km and > 53 km away from UoG TECTC had the odds of 4.50 times (AOR: 4.50; 2.15, 9.40) and 6.02 times (AOR: 6.02; 2.76, 13.14) more likely being late presenter respectively compared to those who resided <3km away from the UoG TECTC.”

Introduction

1. Pg 3, line 14: grammar

Authors’ Response: we accepted and rewritten in the new manuscript.

• Page 3, line 13-14: “Several studies estimated that 10 – 33% of people with glaucoma were visually impaired at their first diagnosis [10-12].”

2. The introduction could make a more clearly argued case for why the study is important: in particular, it cites three studies regarding what are the determinants of late presentation (including one from Africa), but then goes on to say “there is little published evidences [sic] for why some… have advanced disease at first diagnosis”.What are the gaps and/or conflicts in previous work that into this area, and how does the current study hope to address these? in particular, if the authors feel that the previous work from elsewhere in Africa is not applicable to Ethiopia, they should probably state so more explicitly: even better if they can cite supporting demographic / other evidence to support why this is the case. Some of the arguments made in the Discussion regarding weaknesses in previous studies from Africa (e.g. regarding sample size) could be mentioned explicitly in the Introduction.

Authors’ Response: we accepted and rewritten in the new manuscript.

• Page 3, line 22-23 and page 4, line 1-6: “Glaucoma was one of the leading cause of irreversible blindness in Ethiopia, thus this study has an immense importance to salvage the community from glaucoma induced blindness. Ethiopian glaucoma patients have different socio-cultural, genetic, environmental and other population related factors compared to European, Asian and other African countries. Previous published evidences done in Ethiopia doesn’t explore the determinant factors for the severity of glaucoma at the first presentation. Thus, it needs explicit in situ study to identify the determinants for late presentation of glaucoma.”

Materials and Methods

1. pg 5, Line 9: age criterion is missing

Authors’ Response: we accepted and “aged ≥ 18 years” is added in the new manuscript.

2. pg 6, line 21: questionnaire

Authors’ Response: we accepted and spelling is corrected in the new manuscript.

3. pg 7, line 15: “expertise”: experts?

Authors’ Response: we accepted and spelling is corrected as “experts” in the new manuscript.

4. pg 7, line 16: a bit more detail about the reliability check would be useful (assuming it involved retesting, how many people were retested, and what was the interval between testing?): or is this the information that is partly provided on page 8, line 1?

Authors’ Response: we accepted and corrected in the new manuscript.

• Page 7, line 16-20: “The questionnaire was pre-tested for reliability and validity in 25 glaucomatous patients in another hospital (Bahir Dar Felege Hiwot referral eye Hospital) with the same methods and the content of the questionnaire was assessed for its clarity, completeness and modified accordingly. It was also checked for its reliability using a reliability test and has a Cronbach alpha value of 0.77.”

Results

1. It is stated that a history of diabetes was significantly associated with late presentation: isn’t it the ABSENCE of a history of diabetes that is associated?

Authors’ Response: Having of history of diabetes was significantly associated, but “it is protective against late presentation” and we described it in the corrected manuscript:

• Result section: page 15, line 3-5: “On the other hand, those who had history of diabetes mellitus had 84% lesser odds of being late presenter (AOR = 0.16, 95% CI: 0.68, 0.38) compared to those who didn’t have diabetes.”

• We discussed this issues in discussion section: page 19, line 21-23 and page 20, line 1-4: “The present study also revealed that patients who had a history of diabetes mellitus are less likely to present late compared to those who didn’t have diabetes mellitus. The result is comparable with a study done in South Africa [18]. This might be due to the reason that diabetic patients are more likely to have regular medical and ocular examinations for diabetic retinopathy screening and follow-up, hence the opportunity to spot glaucoma at an earlier stage. This can be supported by the evidence of opportunistic detection of glaucomatous optic discs within a diabetic retinopathy screening [31].”

Discussion

1. pg 18, line 20: grammar (sentence fragment)

Authors’ Response: We accepted and rewritten in the new manuscript. 

• Page 18, 20: we corrected and rewritten the sentences.

2. pg 18, line 23: these cited reasons are rather generic: a much better job otargetingspecific reasons is done in the following paragraph when differences between the current study, and the previous South African study, are discussed.

Authors’ Response: We accepted and edited in the new manuscript: Page 18, line 23.

---

## [Decision Letter · Decision Letter 2]

11 Mar 2022

PONE-D-21-27844R2Determinants for late presentation of glaucoma among adult glaucomatous patients in University of Gondar Comprehensive Specialized Hospital. Case-control studyPLOS ONE

Dear Dr. Assefa,

Thank you for submitting your manuscript to PLOS ONE. After careful consideration, we feel that it has merit but does not fully meet PLOS ONE’s publication criteria as it currently stands. Therefore, we invite you to submit a revised version of the manuscript that addresses the points raised during the review process.

We look forward to receiving your revised manuscript.

Kind regards,

David Chau

Academic Editor

PLOS ONE

Journal Requirements:

Reviewers' comments:

Reviewer's Responses to Questions

**Comments to the Author**

1. If the authors have adequately addressed your comments raised in a previous round of review and you feel that this manuscript is now acceptable for publication, you may indicate that here to bypass the “Comments to the Author” section, enter your conflict of interest statement in the “Confidential to Editor” section, and submit your "Accept" recommendation.

Reviewer #4: (No Response)

2. Is the manuscript technically sound, and do the data support the conclusions?

Reviewer #4: Yes

3. Has the statistical analysis been performed appropriately and rigorously? 

Reviewer #4: Yes

4. Have the authors made all data underlying the findings in their manuscript fully available?

Reviewer #4: Yes

5. Is the manuscript presented in an intelligible fashion and written in standard English?

Reviewer #4: No

6. Review Comments to the Author

Reviewer #4: The manuscript has improved in several areas as a result of the authors’ revisions.

However, I find the revision to the Introduction disappointing; it is a superficial rewording, and hasn’t engaged with the criticisms made in my original review appended below:

“The introduction could make a more clearly argued case for why the study is important: in particular, it cites three studies regarding what are the determinants of late presentation (including one from Africa), but then goes on to say “there is little published evidences [sic] for why some… have advanced disease at first diagnosis”. What are the gaps and/or conflicts in previous work that into this area, and how does the current study hope to address these? in particular, if the authors feel that the previous work from elsewhere in Africa is not applicable to Ethiopia, they should probably state so more explicitly: even better if they can cite supporting demographic / other evidence to support why this is the case. Some of the arguments made in the Discussion regarding weaknesses in previous studies from Africa (e.g. regarding sample size) could be mentioned explicitly in the Introduction.”

Several sentences cry out for appropriate references: “Glaucoma is one of the leading causes… [ref?]; Ethiopian patients have [various differences…] [ref?]; previous published evidences done in Ethiopia (sic).. [what evidence? references? Also, “evidence” is already plural - please correct].” At the moment, the modified paragraph is a list of unsupported assertions which weakens the paper substantially, particularly as it is this paragraph that is the one that is supposed to convince the reader there is a need for the current study.

7. PLOS authors have the option to publish the peer review history of their article (what does this mean?). If published, this will include your full peer review and any attached files.

Reviewer #4: No

---

## [Author Response · Author response to Decision Letter 2]

17 Mar 2022

Authors’ response for the Editorials and reviewers’ comment 

Manuscript number: PONE-D-21-27844

Manuscript title: Determinants for late presentation of glaucoma among adult glaucomatous patients in University of Gondar Comprehensive Specialized Hospital: Case-control study

Responses to the Editorials 

1. Journal Requirements:

Authors’ Response: Thank you for your comment! 

• We checked each references and corrected based on the journal reference guidelines.

• We revised the manuscript based on the comments and suggestions of the editor’s and reviewers’.

Responses to the Reviewer #4

1. Reviewer #4: The manuscript has improved in several areas as a result of the authors’ revisions. 

However, I find the revision to the Introduction disappointing; it is a superficial rewording, and hasn’t engaged with the criticisms made in my original review appended below:

“The introduction could make a more clearly argued case for why the study is important: in particular, it cites three studies regarding what are the determinants of late presentation (including one from Africa), but then goes on to say “there is little published evidences [sic] for why some… have advanced disease at first diagnosis”.

What are the gaps and/or conflicts in previous work that into this area, and how does the current study hope to address these? in particular, if the authors feel that the previous work from elsewhere in Africa is not applicable to Ethiopia, they should probably state so more explicitly: even better if they can cite supporting demographic / other evidence to support why this is the case. Some of the arguments made in the Discussion regarding weaknesses in previous studies from Africa (e.g. regarding sample size) could be mentioned explicitly in the Introduction.”

Several sentences cry out for appropriate references: “Glaucoma is one of the leading causes… [ref?]; Ethiopian patients have [various differences…] [ref?]; previous published evidences done in Ethiopia (sic).. [what evidence? references? Also, “evidence” is already plural - please correct].” At the moment, the modified paragraph is a list of unsupported assertions which weakens the paper substantially, particularly as it is this paragraph that is the one that is supposed to convince the reader there is a need for the current study.

Authors’ Response: Thank you very much for your constrictive comment and suggestions. 

• We accepted your comment and revised each of the points you raised in the corrected manuscript. 

• We revised the references and cited statements appropriately. 

• We revised the significance of this study, the gaps of the previous studies and the justifications to conduct this research. 

• We addressed the comments as follows:

• “……….for why some… have advanced disease at first diagnosis”

Corrected in page 3, line 13-16: “Several studies estimated that 10 – 33% of people with glaucoma had advanced disease and visually impaired at the first diagnosis due to their late presentation [10-12]. The reason for the late presentation was due to lack of early symptoms [13, 14], slowly progressive and asymptomatic of nature of glaucoma [15]. 

• “What are the gaps and/or conflicts in previous work that into this area, and how does the current study hope to address these? ..............could be mentioned explicitly in the Introduction.”

Corrected in page 4, line 2-10: “Previous published evidences in Ethiopia didn’t have enough information to explore the determinant factors for the late presentation of glaucoma. Most of these studies were done to assess and the prevalence of blindness due to glaucoma [8, 9] and the associated factors of glaucoma [24]. Because of this, it needs explicit in situ study to identify the determinants for late presentation of glaucoma. In addition, the result of this study will provide base line information for the health care workers, researchers, health care planers, policy makers and other stakeholders accordingly.”

• “Several sentences cry out for appropriate references”

Corrected and cited appropriately, page 4, line 1 and 5: …..Glaucoma was one of the leading cause of irreversible blindness in Ethiopia [8]……... Most of these studies were done to assess and the prevalence of blindness due to glaucoma [8, 9] and the associated factors of glaucoma [24].

---

## [Decision Letter · Decision Letter 3]

12 Apr 2022

Determinants for late presentation of glaucoma among adult glaucomatous patients in University of Gondar Comprehensive Specialized Hospital. Case-control study

PONE-D-21-27844R3

Dear Dr. Assefa,

We’re pleased to inform you that your manuscript has been judged scientifically suitable for publication, pending for some grammatical errors, and will be formally accepted for publication once it meets all outstanding technical requirements.

Kind regards,

David Chau

Academic Editor

PLOS ONE

Additional Editor Comments (optional):

there are still grammatical errors need correction

Reviewers' comments:

Reviewer's Responses to Questions

**Comments to the Author**

1. If the authors have adequately addressed your comments raised in a previous round of review and you feel that this manuscript is now acceptable for publication, you may indicate that here to bypass the “Comments to the Author” section, enter your conflict of interest statement in the “Confidential to Editor” section, and submit your "Accept" recommendation.

Reviewer #2: (No Response)

Reviewer #4: All comments have been addressed

2. Is the manuscript technically sound, and do the data support the conclusions?

Reviewer #2: (No Response)

Reviewer #4: Yes

3. Has the statistical analysis been performed appropriately and rigorously? 

Reviewer #2: (No Response)

Reviewer #4: Yes

4. Have the authors made all data underlying the findings in their manuscript fully available?

Reviewer #2: (No Response)

Reviewer #4: Yes

5. Is the manuscript presented in an intelligible fashion and written in standard English?

Reviewer #2: (No Response)

Reviewer #4: Yes

6. Review Comments to the Author

Reviewer #2: (No Response)

Reviewer #4: (No Response)

7. PLOS authors have the option to publish the peer review history of their article (what does this mean?). If published, this will include your full peer review and any attached files.

Reviewer #2: **Yes: **Bing Jiang

Reviewer #4: No

---

## [Editor Report · Acceptance letter]

22 Apr 2022

PONE-D-21-27844R3 

Determinants for late presentation of glaucoma among adult glaucomatous patients in University of Gondar Comprehensive Specialized Hospital. *Case-control study*

Dear Dr. Assefa:

I'm pleased to inform you that your manuscript has been deemed suitable for publication in PLOS ONE. Congratulations! Your manuscript is now with our production department. 

Kind regards, 

on behalf of

Dr. David Chau 

Academic Editor

PLOS ONE